# Coumarin Ketoxime Ester with Electron-Donating Substituents as Photoinitiators and Photosensitizers for Photopolymerization upon UV-Vis LED Irradiation

**DOI:** 10.3390/polym14214588

**Published:** 2022-10-28

**Authors:** Shuheng Fan, Xun Sun, Xianglong He, Yulian Pang, Yangyang Xin, Yanhua Ding, Yingquan Zou

**Affiliations:** 1College of Chemistry, Beijing Normal University, No. 19, Xinjiekouwai St. Haidian District, Beijing 100875, China; 2Hubei Gurun Technology Co., Ltd., Jingmen Chemical Recycling Industrial Park, Jingmen 448000, China

**Keywords:** photoinduced polymerization, photoinitiator, oxime ester, LED

## Abstract

High-performance photoinitiators (PIs) are essential for ultraviolet–visible (UV-Vis) light emitting diode (LED) photopolymerization. In this study, a series of coumarin ketoxime esters (COXEs) with electron-donating substituents (*tert*-butyl, methoxy, dimethylamino and methylthio) were synthesized to study the structure/reactivity/efficiency relationships for substituents for the photoinitiation performance of PIs. The introduction of heteroatom electron-donating substituents leads to a redshift in the COXE absorption of more than 60 nm, which matches the UV-Vis LED emission spectra. The PIs also show acceptable thermal stability via differential scanning calorimetry (DSC) and thermal gravimetric analysis (TGA). The results from real-time Fourier transform infrared (RT-FTIR) measurements indicate that COXEs show an excellent photoinitiation efficiency for free radical polymerization under UV-Vis LED irradiation (365–450 nm); in particular, the conversion efficiency for tri-(propylene glycol) diacrylate (TPGDA) polymerization initiated by COXE-O and COXE-S (4.8 × 10^−5^ mol·g^−1^) in 3 s can reach more than 85% under UV-LED irradiation (365, 385 nm). Moreover, the photosensitization of COXEs in the iodonium hexafluorophosphate (Iod-PF_6_) and hexaarylbiimidazole/N-phenylglycine (BCIM/NPG) systems was investigated via RT-FTIR. As a coinitiator, COXEs show excellent performance in dry film photoresist (DFR) photolithography. This excellent performance of COXEs demonstrates great potential for UV-curing and photoresist applications, providing a new idea for the design of PIs.

## 1. Introduction

Recently, photopolymerization technology has developed rapidly and is widely used in various fields [1,2,3], such as the conventional photocuring industry (coating, paint and ink) [4], lithography technology, photosensitive imaging materials [5], adhesives, integrated circuits and optical elements [6], 3D and 4D printing [7,8,9,10], dental materials [11,12], photovoltaics [13], multifunctional lenses and hydrogels [14,15]. It is a “5E” technology that is summarized as having “economical, environmentally friendly, energy saving, enabling and efficient characteristics [16,17]”.

UV-Vis LED photopolymerization has attracted increasing attention from academic institutes and industry sectors with the global implementation of the “Minamata Convention on Mercury” to protect the human environment and health from the harm of mercury [18,19]. UV-Vis LED photopolymerization not only exhibits better performance than traditional photopolymerization, such as mild reaction conditions at room temperature, fast reaction completion in seconds or minutes, and low VOC release [6,17], but also shows the following advantages: monochromatic light (typically 20 nm), high light energy absorption, low energy consumption, long service life, no ozone release, easy and safe operation and programmable control [6,17]. However, the narrow emission bandwidth of LED lamps and the limit of LED technology development enables the LED lamp operation at a relatively long emission wavelength in the UV-A and visible light bands [20,21], which limits the use of traditional commercialized PIs (the absorption usually occurs at the UV-C and UV-B bands [22]) upon LED exposure. Therefore, it is necessary to develop new PIs suitable for the UV-Vis LED emission spectrum.

In recent years, a series of works on PIs suitable for LEDs have been reported, such as thioxanthone derivatives [22], furan derivatives [23], coumarin derivatives [24,25], indane-1,3-dione derivatives [26] and carbazole-based PIs [3]. These PIs usually contain a chromophore structure and are used in multicomponent photoinitiation systems (Norrish type II PIs). Germanium-based PIs have a high initiation performance under LED light [27], but their synthesis process is complex and expensive, limiting their commercial value.

Oxime esters are efficient photoinitiators because the N-O bond of the structure can be used to generate free radicals quickly and efficiently upon light exposure [28,29,30]. Oxime esters with long wavelength absorption have attracted general interest [20,31,32,33,34,35]. To extend the absorption wavelength range to the UV-A or even visible light, it is a common choice to introduce excellent chromophores into oxime esters. Coumarins have been proven to be potential candidates. As an effective fluorescent heterocycle, coumarin has high electron-transfer quantum yields in excited states, making it a versatile chromophore with an electron donor or acceptor [24,36]. Because of these excellent photophysical properties, coumarin has various applications in photopolymerization, such as for fluorescent molecular probes [37], functional monomers and polymers [38,39], sensitizers and initiators [40,41,42]. However, the reported coumarin-based sensitizers or oxime esters mostly focus on the 7-position substitute coumarin chromophores, such as 7-diethylaminocoumarin or 7-methoxycoumarin, which exhibit excellent photosensitive performance [20,31,43]. However, there are other substitution positions of the coumarin chromophore, such as the 6-position substitution (see Figure 1), which also deserve a thorough investigation.

In this work, 6-position electron donors and 8-position *tert*-butyl substituted coumarin ketoxime esters (COXEs) were first designed and synthesized by introducing *tert*-butyl, methoxy, dimethylamino and methylthio electron-donor group into the coumarin (see Figure 2), respectively. The effect of the heteroatom electron-donor group at the 6-position substitution of coumarin on the structure-efficiency relationship was investigated via a series of measurements, including UV-Vis absorption spectroscopy, real-time Fourier transform infrared (RT-FTIR) spectroscopy upon UV-Vis LED irradiation (365–450 nm), fluorescence emission spectroscopy, etc. The chemical mechanism was investigated via steady-state photolysis experiments, electron spin resonance (ESR) and LC-MS. Moreover, the photosensitive performance of oxime esters to N-phenyl-glycine (NPG) and hexaarylbiimidazole (BCIM) was reported first, which may enable oxime esters to be used in many more practical applications, such as dry film photoresists (DFRs).

## 2. Materials and Methods

### 2.1. Synthesis

All COXES were synthesized via a 4-step reaction from ortho-para-substituted phenol (Figure 2). These include the Friedel–Crafts acylation reaction [44], knoevenagel condensation reaction, oxidation reaction [45] and general esterification reaction. In the whole synthesis process, the yield of each step reached 80%. Characterization data are shown in the Supporting Information.

3-[(1-[(acetyloxy)imino]ethyl]-6,8-bis(1,1-dimethylethyl)-2H-1-benzopyran-2-one (COXE-C). NMR δ (ppm) ^1^H-NMR (600 MHz, CDCl_3_) δ 8.06 (s, 1H), 7.60 (d, *J* = 6 Hz, 1H), 7.35 (d, *J* = 6 Hz, 1H), 2.41 (s, 3H), 2.25 (s, 3H), 1.50 (s, 9H), 1.33 (s, 9H). 13C NMR (150 MHz, CDCl_3_) δ168.21, 161.82, 158.64, 150.03, 148.28, 143.62, 132.79, 132.69, 132.13, 129.55, 128.26, 124.80, 123.61, 118.34, 34.70, 31.15, 19.68, 15.78. HRMS (MALDI-TOF) *m*/*z* calcd for C_21_H_27_NO_4_ [M+H]^+^ 358.2013, found 358.2010.

3-[(1-[(acetyloxy)imino]ethyl]-8-(1,1-dimethylethyl)-6-methoxy-2H-1-benzopyran-2-one (COXE-O). NMR δ (ppm) ^1^H-NMR (600 MHz, CDCl3) δ 8.02 (s, 1H), 7.16 (d, *J* = 6 Hz, 1H), 6.78 (d, *J* = 6 Hz, 1H), 3.82 (s, 3H), 2.41 (s, 3H), 2.25 (s, 3H), 1.48 (s, 9H). ^13^C NMR (150 MHz, CDCl3) δ 168.27, 162.03, 158.94, 155.78, 148.00, 143.95, 139.78, 122.82, 119.98, 119.29, 107.65, 55.77, 35.12, 29.77, 19.70, 15.81. HRMS (MALDI-TOF) *m*/*z* calcd for C_18_H_21_NO_5_ [M+H]^+^ 332.1492, found 332.1503.

3-[(1-[(acetyloxy)imino]ethyl]-8-(1,1-dimethylethyl)-6-dimethylamino-2*H*-1-benzopyran-2-one (COXE-N). NMR δ (ppm) ^1^H-NMR (600 MHz, CDCl_3_) δ 8.00 (s, 1H), 7.06 (s, 1H), 6.61 (s, 1H), 2.96 (s, 6H), 2.41 (s, 3H), 2.24 (s, 3H), 1.49 (s, 9H). ^13^C NMR (150 MHz, CDCl_3_) δ 168.36, 162.29, 159.24, 147.29, 145.67, 144.55, 138.34, 122.47, 119.47, 117.04, 108.42, 41.02, 35.22, 29.88, 19.72, 15.85. HRMS (MALDI-TOF) *m*/*z* calcd for C_19_H_24_N_2_O_4_ [M+H]^+^ 345.1809, found 345.1812.

3-[(1-[(acetyloxy)imino]ethyl]-8-(1,1-dimethylethyl)-6-dmethylthio-2H-1-Benzopyran-2-one (COXE-S). NMR δ (ppm) ^1^H-NMR (600 MHz, CDCl_3_) δ 8.01 (s, 1H), 7.45 (d, *J* = 6 Hz, 1H), 7.21 (d, *J* = 6 Hz, 1H), 2.50 (s, 3H), 2.41 (s, 3H), 2.25 (s, 3H), 1.48 (s, 9H). ^13^C NMR (150 MHz, CDCl_3_) δ 168.23, 161.80, 158.60, 151.23, 143.44, 138.61, 134.66, 129.99, 124.20, 123.12, 119.59, 35.21, 29.78, 19.69, 16.59, 15.80. HRMS (MALDI-TOF) *m*/*z* calcd for C_18_H_21_NO_4_S [M+H]^+^ 348.1264, found 348.1254.

### 2.2. Materials

2-(1,1-dimethylethyl)-4-methoxy-phenol, 2-(1,1-dimethylethyl)-4-dimethylamino-phenol, 2-(1,1-dimethylethyl)-4-methylthio-phenol and 2,4-(1,1-dimethylethyl)-phenol were purchased from Shanghai Bide Pharmatech Co., Ltd. (Shanghai, China). Commercialized oxime ester PIs OXE-01, OXE-02 and OXE-03 (Figure 1) were purchased from BASF (China) Co., Ltd. (Shanghai, China). Tri-(propylene glycol) diacrylate (TPGDA) (Figure 1) was purchased from Shanghai Yinchang New Material Co., Ltd. (Shanghai, China). Iodonium hexafluorophosphate (Iod-PF_6_) (Figure 1) was obtained from Hubei Gurun Technology Co., Ltd. (Jingmen, China). Bis(2-chlorophenyl)-tetraphenyl biimidazole (BCIM), *N*-phenylglycine (NPG), *N*-*tert*-butyl-2-phenylnitrone (PBN) and *tert*-butylbenzene were purchased from Beijing Innochem Science & Technology Co., Ltd. (Beijing, China). All materials were obtained from commercial sources and used without further treatment.

### 2.3. Characterization

NMR spectra were recorded using a JOEL 600 MHz, and deuterated chloroform was used as the solvent. Mass spectra (MALDI) were acquired using a FT-ICR spectrometer (Bruker Daltonics Inc. BIFLEX III), acetonitrile was used as the solvent. UV-Vis absorption spectra and steady-state photolysis tests for COXEs were measured using a Shimadzu UV 3600, acetonitrile was used as the solvent and the PI concentration was 50 ppm. Fluorescence spectra were measured using a FLS980, acetonitrile was used as the solvent, and PI concentration was 50 ppm. LC-MS measurements were carried out by high-performance liquid chromatography (Agilent 1260) and quadrupole time-of-flight tandem mass spectrometry (Bruker micrOTOF-QII). The chromatographic separation was performed using a BetaBasic-18 column eluted with a mixture composed of 90% acetonitrile and 10% purified water at a flow rate of 0.3 mL·min^−1^; the monitoring wavelength was 254 nm; acetonitrile was used as the solvent.

### 2.4. Excited State Lifetime Test

Single excited state lifetime (fluorescence lifetime) and triplet excited state lifetime (phosphorescence lifetime) tests were recorded using an Edinburgh FLS980 fluorescence spectrometer with a microsecond flash lamp (uF900). The fluorescence and phosphorescence lifetimes (τ) of solid samples were determined by fitting the decay curve with a multiexponential decay function of I(*t*) = A_1_exp(−*t*/τ_1_) + A_2_exp(−*t*/τ_2_) +…+ A_i_exp(−t/τ_i_), where A_i_ and τ_i_ represent the amplitudes and lifetimes of the individual components of the multiexponential decay profiles [46].

### 2.5. Theoretical Calculations

Theoretical calculations of COXEs were computed by using the Gaussian 09W package (based on density functional theory) with utilization of the Chemcraft software for drawing and visual analysis. The optimized molecular geometries of the ground state, the molecular transition method and frontier molecular orbit (HOMO and LUMO) of COXEs were obtained at the TD/B3LYP/6-31G(d) level (isovalue = 0.03).

### 2.6. Solubility Measurements

COXEs (0.5 g) were added to 3 g monomer TPGDA each time until COXEs could no longer be dissolved, and the ultrasonic bath was maintained during this period. The supernatant (one drop) was taken as a saturated solution by centrifugation and diluted with acetonitrile. The absorbance of the supernatants was measured via UV-Vis absorption spectroscopy. The solubility of COXEs in TPGDA was determined quantitatively based on the relationship between the measured absorbance and the known molar extinction coefficient of COXEs [47].

### 2.7. ESR Experiments

ESR experiments for COXEs were carried out by using a JEOL JES-FA200 spectrometer (X-band) at a field modulation frequency of 9.06 GHz and 100 kHz, and the microwave power was 0.998 mW. The radicals were generated at room temperature upon LED irradiation at 400 nm under a nitrogen atmosphere and then trapped by PBN. COXEs and PBN were dissolved in *tert*-butylbenzene, and the COXE concentration was 3 × 10^−4^ mol·L^−1^. The COXE to PBN molar ratio was 1:5. The ESR spectrum simulation was conducted using MATLAB software.

### 2.8. Thermal Stability Measurements

Differential scanning calorimetry (DSC) measurements for COXEs mixed with TPGDA (COXE concentration of 4.8 × 10^−5^ mol·g^−1^) were carried out by METTLER TOLEDO. The mixture was heated from 25 °C to 250 °C at a rate of 10 K/min under nitrogen atmospheric conditions. The COXE concentration was 4.8 × 10^−5^ mol·g^−1^ resin. Thermal gravimetric analysis (TGA) measurements for COXEs were performed by using a Mettler Toledo TGA 1/1100SF with approximately 5 mg of the sample under a nitrogen atmosphere. The temperature was ramped from 25 to 500 °C at a rate of 10 K/min.

### 2.9. RT-FTIR Measurements

The study of photopolymerization kinetics is based on RT-FTIR measurements. The prepared photoinitiation formulas contained TPGDA and COXEs. The concentration of COXEs in TPGDA was 4.8 × 10^−^^5^ mol·g^−1^ based on 2 wt% OXE-02 or 2.4 × 10^−^^6^ mol·g^−1^ based on 0.1 wt% OXE-02. The formulas were spread evenly and laminated between two KBr plates and then scanned by a RT-FTIR (Nicolet instrument, Thermo Fisher Scientific Inc., Waltham, MA, USA). Five LED light sources (365, 385, 400, 425 and 450 nm) were used in this study, and the light intensity was kept at 100 mW·cm^−2^. Commercialized oxime ester PI OXE-02 and newly developed longer wavelength oxime ester PI OXE-03 were both selected as references. The conversion of TPGDA polymerization initiated by different concentrations of COXEs or Iod-PF_6_/COXE, COXE/NPG and BCIM/NPG/COXE systems, was measured in the same way. The C=C double bond conversion of TPGDA was calculated using the following equation:(1)$$\mathrm{Conversion}\left(\%\right)=\frac{A_{0}-A_{t}}{A_{0}}\times 100\;\%$$
where A_0_ is the peak area in the stretching vibration characteristic region of the C=C double bond before exposure (1653−1603 cm^−1^ for TPGDA), and A_t_ is the peak area after exposure time t.

### 2.10. Gibbs Free Energy Changes

Gibbs free energy changes (ΔG) for the electron transfer reaction between COXEs as electron donors and Iod-PF_6_ as an electron acceptor, or COXEs as electron acceptors and NPG as an electron donor were calculated from the Rehm–Weller equation [48]:ΔG = E_ox_ − E_red_ − E_s1_ + C(2)
where E_ox_ is the oxidation potential of the electron donors and E_red_ is the reduction potential of the electron acceptor. Es_1_ is the excited singlet state energy of COXEs determined from the maximum UV–Vis absorption, and C is the electrostatic interaction energy, which is usually negligible in polar solvents. E_ox_ and E_red_ were measured by cyclic voltammetry in acetonitrile (10^−3^ mol/L). Tetrabutylammonium hexafluorophosphate (0.1 mol·L^−1^) was used as a supporting electrolyte. Ferrocene was used as a reference standard, a platinum disc was used as a working electrode and Ag/AgCl was used as a reference electrode.

### 2.11. Photolithography Performance Measurements for DFRs

The prepared DFR formulas contained a polyethylene terephthalate film, a protective polyethylene film, a photosensitive resin [49], bisphenol-A-ethoxylate dimethacrylate as the monomer, BCIM (3 wt%) as the photoinitiator and COXEs (0.1 wt%) as the coinitiator.

The photolithography procedure for the prepared DFRs is shown in Figure 2. The films were developed under 405 nm LED irradiation (energy 10 mJ·cm^−2^ ) for 15 min with a 405 nm mask aligner. The resolution and adhesion areas were imaged with an optical microscope (12XB-PC, Shanghai Optical Instrument Factory (Shanghai, China)) at 200× magnification and defined as the minimum “Line/Space (L/S)” value. The resolution, color difference value ΔE_ab_ and photosensitivity were measured.

## 3. Results and Discussion

### 3.1. Solubility of COXEs

The introduction of *tert*-butyl groups on the coumarin chromophore greatly increases the solubility. *Tert*-butyl groups can increase the spatial stereoscopic property of the structure and decrease the intermolecular interactions [50], which makes the raw structure with planarity easier to solvated, and the aggregation reduced, so the dissolution rate in the monomer can also be improved. The results of the solubility measurements show that the maximum solubility (g/100 g) in acrylic monomer TPGDA is increased from 0.242 (*tert*-butyl-unsubstituted, see Appendix A) to 4.34 for COXE-O (*tert*-butyl-monosubstituted) and 10.3 for COXE-C (*tert*-butyls-disubstituted). The solubility is increased by 43 times, which is beneficial for the practical application of this kind of PI. The optimized horizontal structures of COXEs are shown in Appendix A.

### 3.2. Photophysical Properties of COXEs

The photophysical properties, including UV-Vis absorption properties, ground-state electronic properties and cleavage excited states, are essential for the design and application of PIs. Because of the introduction of an electron donor, compared with COXE-C, the other three compounds, COXE-O, COXE-S and COXE-N, lead to the formation of a new absorption peak; their absorption is significantly redshifted (60 nm, 70 nm and 100 nm, respectively), and the maximum absorption of COXE-N can reach 530 nm. In addition, the greater the redshift, the lower the intensity of the maximum absorption peak. The UV-Vis absorption curves for COXEs are shown in Figure 3, and the data are shown in Table 1. The detailed molar extinction coefficients measured at LED emission wavelengths (365, 385, 400, 425 and 450 nm) are shown in Appendix A.

Electron-donating conjugation on the chromophore can also impact the fluorescence spectrum. The fluorescence emission from COXE-O, COXE-S and COXE-N is significantly redshifted (40 nm, 70 nm and 180 nm, respectively) compared to that for COXE-C, as shown in Figure 3 and Table 1. The Stokes shift is less than that for COXE-C, which indicates that the introduction of methoxy, methylthio and dimethylamino substituents weakens the change in the PI geometry between the ground state and first excited singlet state [20]. In addition, only COXE-N shows the lowest fluorescence quantum yield (Φ_f_), which is 0.6%, and a particularly long fluorescence lifetime of 1512.90 ns. This result indicates that the introduction of dimethylamino to COXE leads to more nonradiative deactivation, which is supported by the short energy gap between S_0_-S_1_ for COXE-N (see Figure 4) [35], and decreases the amount of PI transformed from the excited singlet into the excited triplet via ISC [34,51].

The aforementioned photophysical properties are influenced by the properties of frontier orbitals. Figure 4 shows that the introduction of heteroatom electron-donating substituents (methoxy, methylthio and dimethylamino) strengthens the delocalization of the highest occupied molecular orbital (HOMO) and leads to the formation of a π-MO located at the heteroatom electron-donating substituents and coumarin ring. The HOMO-LUMO energy gap (ΔE_HOMO-LUMO_) is also reduced, which leads to an increased maximum absorption wavelength for COXEs [52].

The time-dependent density functional theory (TD-DFT) calculations were performed to calculate the energy of the ground states (S_0_) and excited states (S_1_ and S_2_) of COXEs (Table 2), including simulated absorption wavelengths (π_abs_), oscillator strength (*f*), transition types and the main transition assignments. The results show that the S_0_→S_1_ transition for COXEs occurs from the HOMO to LUMO, and the S_0_→S_2_ transition for COXEs occurs from the HOMO_−1_ to LUMO. Furthermore, the S_0_→S_1_ and S_0_→S_2_ transitions are driven by π→π* transitions from the conjugated system of the substituted coumarin ring to the conjugated system of the C = N bond, leading to long-range electron delocalization from the coumarin subunit to the oxime ester (see Figure 4). According to the oscillator strength (*f*), the lowest energy absorption bands for COXE-O, COXE-N and COXE-S all include a weaker excited singlet transition S_1_ and a stronger S_2_. COXE-C shows a small difference in the oscillator strengths for S_1_ and S_2_, which indicates that the introduction of heteroatom electron-donating substituents (methoxy, methylthio and dimethylamino) lead to a decrease in the oscillator strength of S_1_ despite a significant redshift in the absorption wavelength. Therefore, the molar extinction coefficients at maximum absorption for COXE-O, COXE-N and COXE-S are lower than that for COXE-C (see Table 1).

### 3.3. Photochemistry Properties of COXEs

#### 3.3.1. ESR Experiment

Oxime esters have been reported to generate iminyl and acyloxy radicals via the cleavage of N-O bonds under exposure [20,31,53,54]. ESR experiments were conducted to confirm the kinds of free radicals generated by COXEs upon UV LED irradiation. The results are shown in Figure 5, which indicate that the same radical is generated (acetoxy radical (*α*_N_(G) = 13.48, *α*_H_(G) = 1.95) as that trapped by PBN [55,56]. No hyperfine coupling constant for methyl radicals is found in the ESR spectrum. The clear ESR spectrum further proves the first cleavage of the N-O bond after exposure, as shown in Figure 3. In addition, the ESR signals for COXE-N are not obvious, thus, indicating that dimethylamino-substituted COXE-N undergoes slow cleavage. This conclusion is also reflected in the steady-state photolysis test and RT-FTIR measurement discussed below.

#### 3.3.2. Steady-State Photolysis Analysis

The steady-state photolysis analysis was performed using a UV-Vis spectrophotometer equipped with a 400 nm LED to observe the photostability of COXEs in acetonitrile. The steady-state photolysis curves shown in Figure 6 show that the introduction of methoxy and methylthio accelerates the decrease in the absorption peak and leads to a slight blueshift (2 nm) following irradiation with a 400 nm LED. Compared to COXE-C, which decreased by only 9.5% within 30 min, COXE-S decreased most significantly by 53.2% (294 nm) and 43.7% (369 nm), and COXE-O decreased by 37.0% (292 nm) and 30.2% (358 nm) within 30 min. This indicates that COXE-S and COXE-O have a higher photoreactivity at 400 nm. Meanwhile, the absorption of COXEs between 405 nm and 475 nm is slightly increased with increasing irradiation time, probably because the coupling reaction of two coumarin-iminyl radicals occurs, and new compounds are formed by generating N-N bonds (see Figure 2). This compound (M = 544.23) can be found in the LC signals (see Appendix A). The photolysis of COXE-N begins almost after 5 min, further demonstrating that COXE-N has a slow cleavage rate.

#### 3.3.3. Proposed Photolysis Mechanism

Different LC signals and similar mass spectra obtained for COXE-O by LC-MS measurements also show that a part of the COXE-O-formed isomers are not cleaved after exposure but undergo cis–trans isomerization on the N-O bond [57]. The detection of CO_2_ by the IR spectrum peak of 2337 cm^−1^ and photodecarboxylation experiments are shown in Appendix A, which illustrate the further cleavage of CO_2_ from acetoxy radicals to form methyl radicals that have a powerful ability to initiate polymerization. Based on the above experimental results, the photolysis mechanism for COXEs represented by COXE-O was speculated and is shown in Figure 3.

### 3.4. Thermal Stability of COXEs

Thermal stability is a major issue in the development of oxime ester PIs because of the low bond energy of N-O, which is approximately 228 kJ/mol [43]. The thermodynamic stability and storage stability of COXEs in the formula were measured to evaluate their application potential. First, the initiation temperature of COXEs in TPGDA was measured by DSC. The results are shown in Appendix A. All COXEs have a similar initiation temperature of approximately 130 °C, which is significantly higher than that for commercialized OXE-01 at 90 °C and slightly different from that for OXE-02 and OXE-03. Furthermore, the thermal cracking temperature of these PIs in the solid-state was also studied via TGA. The results shown in Appendix A show that all decomposition temperatures for COXEs are higher than 170 °C, and that for COXE-C, COXE-O and COXE-N are higher than that for OXE-01. These results indicate that the thermal stability of COXEs is higher than that of OXE-01 and is sufficient for daily and usual storage.

### 3.5. Photopolymerization Kinetics for COXEs as Photoinitiators

In this section, the photopolymerization kinetics for COXEs are studied via RT-FTIR, and two commercial oxime esters, OXE-02 and OXE-03, are selected as references.

#### 3.5.1. Effect of Chemical Structure on Photopolymerization Kinetics

Different UV-Vis LEDs emitting in the wavelength range from 365 nm to 450 nm were employed as light sources to study the effect of chemical structure on the photopolymerization kinetics of COXEs, while TPGDA was used as a monomer. The results are shown in Figure 7 and Appendix A and Table 3. Upon UV-LED emission at 385 nm irradiation, the conversion of TPGDA polymerization initiated by COXE-O and COXE-S in 3 s can reach 85%, more than 10% higher than that of OXE-03, which indicates that the photoinitiation performance of COXE-O and COXE-S are both better than OXE-03 under UV-LED irradiation (see Figure 7a and Table 3).

Under 400 nm LED irradiation, the conversion of TPGDA polymerization initiated by COXE-O (83%) and COXE-S (92%) in 10 s reaches a value of more than 60% higher than that of OXE-02 (21%) (Figure 7b and Table 3). Under 450 nm irradiation, the conversion initiated by OXE-03 is only 3.26%, and OXE-02 fails to initiate after 20 s of exposure (Figure 7c and Table 3). This indicates that the photoinitiation properties of COXE-O and COXE-S are significantly better than those of OXE-02 under visible LED irradiation and OXE-03 under 450 nm irradiation. One of the reasons for this phenomenon is that the absorption wavelengths for COXE-O and COXE-S are extended to the visible region, which is a better match for the vis-LED light sources compared to OXE-02 and OXE-03.

Comparing the four kinds of COXEs, the introduction of methoxy and methylthio results in COXE-O and COXE-S having a significantly better photoinitiation efficiency than COXE-C, as shown in Figure 7. COXE-S is better than COXE-O in the visible region. Under 365 nm irradiation, the conversion of TPGDA polymerization initiated by COXE-C can still reach 82% in 3 s, but decreases to only 23% under 385 nm irradiation and fails to initiate polymerization under 450 nm irradiation, which may explain the short absorption wavelength of COXE-C (see Figure 3a). However, COXE-N has a maximum absorption wavelength of 510 nm, and the molar extinction coefficient of COXE-N is nearly 40 times that of COXE-O at 450 nm, but the photoinitiation performance of COXE-N is the worst. The introduction of dimethylamino into the 6-position substituted for coumarin leads to a long fluorescence lifetime and very low fluorescence yields meaning to undergo more non-radiation deactivation, which can explain the decrease in the cleavage efficiency for COXE-N (see Table 1). Therefore, the photoinitiation performance cannot be assessed based on only the molar extinction coefficient and absorption wavelength.

#### 3.5.2. Effect of PI Concentration on Photopolymerization Kinetics

The effect of the PI’s concentration on photopolymerization kinetics was studied using COXE-O and COXE-S as PIs. The results show that with the increasing PI concentration, the conversion of TPGDA is increased. In particular, when the PI concentration is increased from 2.4 × 10^−6^ mol·g^−1^ to 1.2 × 10^−5^ mol·g^−1^, both the final conversion of TPGDA and the polymerization rate are significantly improved, as shown in Figure 8. However, when the concentration of COXEs is increased from 1.2 × 10^−5^ mol·g^−1^ to 2.4 × 10^−5^ mol·g^−1^, a slight change for the case of COXE-O, nearly no change for the case of COXE-S is observed because of the cage effect [58]. It is necessary to determine the proper concentration of PI in practical applications.

### 3.6. Photopolymerization Kinetics of COXEs as Photosensitizers

#### 3.6.1. Photopolymerization Kinetics of Iod-PF_6_/COXE Systems

We were surprised to find that a low concentration of COXEs can be used to sensitize the low concentration of iodonium salt to initiate the polymerization of TPGDA. As shown in Figure 9, when the concentration of COXEs is 2.4 × 10^−6^ mol·g^−1^, the polymerization of TPGDA cannot be well initiated in 100 s upon 400 nm LED irradiation. The photoinitiation efficiency of a single Iod-PF_6_ was measured under the same conditions, and the results show that a single Iod-PF_6_ almost fails to initiate TPGDA photopolymerization under 400 nm irradiation in 100 s. However, when we mix the COXEs with Iod-PF_6_ together, the coinitiator system shows excellent performance to initiate the polymerization of TPGDA, as shown in Figure 9b. Except for COXE-C, the polymerization conversion of TPGDA initiated by other COXEs/Iod-PF_6_ systems is similar, reaching a value of nearly 80% in 60 s. This result indicates that the introduction of heteroatom electron-donating substituents (methoxy, dimethylamino and methylthio) can improve the sensitization efficiency of COXEs.

Furthermore, the effects of the concentrations of COXEs and Iod-PF_6_ in two-component photoinitiation systems were also investigated. The molar mass of Iod-PF_6_ (6.0 × 10^−5^ mol·g^−1^) in TPGDA was maintained, and the proportion of COXE-O was changed. Figure 10a indicates that only a small molar mass proportion of COXE-O (0.02) is required to significantly improve the photoinitiation efficiency of Iod-PF_6_, and the initiation efficiency continues to increase with increasing COXE-O. Additionally, the molar mass of COXE-O (2.4 × 10^−6^ mol·g^−1^) in TPGDA was maintained, and the proportion of Iod-PF_6_ was changed. Figure 10b shows that the photoinitiation efficiency of TPGDA polymerization is also increased with the increasing Iod-PF_6_ concentration. Therefore, for the coinitiator system of COXEs and Iod-PF_6_, it is necessary to find the proper concentration of both components in practical applications.

#### 3.6.2. Photopolymerization Kinetics of BCIM/NPG/COXE Systems

Hexaarylbiimidazole (HABI) derivatives, such as BCIM, are commonly used as PIs in the industrial production of DFRs [59]. However, because of their short-wavelength absorption, which is below 350 nm, and as one kind of type II PI, this kind of compound is usually used together with hydrogen donors such as NPG and longer wavelength photosensitizers such as 4,4′-bis(dimethylamino)benzophenone [60,61,62]. In this section, we found that COXEs can be used to improve the photoreactivity of the BCIM/NPG coinitiator system upon 450 nm LED irradiation. Moreover, the performance of COXEs can be improved by using NPG and longer wavelength photosensitizers such as 4,4′-bis(dimethylamino)benzophenone [62]. Compared with Figure 11a,b, a significant synergistic effect of COXEs and BCIM/NPG can be found. The COXE-initiating system does not exhibit excellent photoinitiating performance at such low concentrations, but when the same concentration of COXEs was added to the BCIM/NPG system, the final conversion of TPGDA was increased compared with that for the system without COXEs, and the photopolymerization rate was greatly improved compared with that for the COXE system. The conversion of TPGDA polymerization initiated by the BCIM/NPG/COXE three-component system can reach more than 80% in 60 s, which is 10% higher than that for the BCIM/NPG two-component system. Surprisingly, we also found that the appearance of NPG can improve the photoinitiating performance of COXEs, as shown in Figure 11c. The hydrogen donor additive NPG plays a critical role in improving the photoinitiation performance of the BCIM system and COXEs because it can generate another reactive aminoalkyl radical (see Figure 4) [63]. A similar measurement result is obtained under 400 nm irradiation (Appendix A).

#### 3.6.3. Sensitization Mechanisms for COXEs

The above studies prove the sensitization of COXEs in Iod-PF_6_ and BCIM/NPG systems, which can be attributed to the photoinduced electron transfer (PET) reaction between COXEs and Iod-PF_6_ or NPG, which can be evaluated from the Gibbs free energy change (ΔG_et_) < 0 [54]. To confirm this, cyclic voltammetry was conducted to measure the redox potentials (E_ox_, E_red_) for COXEs, and ΔG_et_ was calculated via Equation (1). Table 4 shows that the ΔG_et_ for all Iod-PF_6_/COXEs or COXEs/NPG systems are negative, which indicates that the PET reaction is thermodynamically favorable from COXEs to Iod-PF_6_ (see r1 and r2) or from NPG to COXEs (see r3 and r4). However, the ΔG_et_ between BCIM and COXEs is positive, which means that it is difficult for PET to occur from COXEs to BCIM. Therefore, for the three-component system of COXEs/BCIM/NPG, a synergistic effect occurs between the COXEs/NPG coinitiator system and the BCIM/NPG coinitiator system when they initiate the photopolymerization of TPGDA upon 450 nm LED exposure. In addition, the higher the HOMO energy of COXEs, the easier it is to transfer electrons to deoxidize Iod-PF_6_ [64], which can explain the difference in the sensitization efficiency of COXEs observed in Figure 9b.

ESR experiments were conducted to investigate the free radicals generated by Iod-PF_6_/COXEs and COXEs/NPG. The ESR signals trapped by PBN are shown in Figure 12. The hyperfine coupling constant of the aryl radical [24] (*α*_N_(G) = 14.06, *α*_H_(G) = 2.14) was obtained from Iod-PF_6_/COXEs, which proves the existence of the radical active species Ar^•^ in the Iod-PF_6_/COXE systems under LED irradiation. The photochemical mechanism of the Iod-PF6/COXE interaction is proposed in reactions(r1) and (r2) in Figure 4.

ESR signals for COXEs/NPG systems show the existence of a NPG_(-H,CO2)_^•^ radical. It can be considered that a photoinduced electron transfer reaction in COXEs/NPG systems can lead to the generation of a NPG_(-H)_^•^ radical, and that a decarboxylation reaction occurs to transform NPG_(-H)_^•^ to NPG _(-H,CO2)_^•^ [24,25]. NPG _(-H,CO2)_^•^ is an active specie for initiating polymerization. The photochemical mechanism for the COXEs/NPG interaction is proposed in Reactions (r3) and (r4) in Figure 5.

#### 3.6.4. Photolithography of Dry Film Photoresist Using COXEs as a Coinitiator

Photoresist materials are a key component of photolithography technologies [67], and the PI HABI system plays a critical role in photoresist materials in terms of photosensitivity and resolution [68]. In this section, we applied the BCIM/NPG/COXE coinitiator system for the manufacture of DFRs and tested the performance of the DFRs, which can be used in practical applications. In the DFR test without COXEs, it was found that the color difference (ΔE_ab_) was 4.7, which could not reach the application standard. After adding COXE-O or COXE-S as the coinitiator, respectively, the color difference value in the DFR test was significantly improved, so the test data were subsequently supplemented. The results are shown in Figure 13, which show the optical microscopy images of the photolithographic patterns formed via the exposure and development of the DFRs. The performance data for DFRs containing photosensitivity, color difference value ΔE_ab_ and resolution of the resolution area and adhesion area are summarized in Table 5. The results indicate that the DFRs using COXE-O and COXE-S show an acceptable ΔE_ab_ and resolution in the resolution region. This finding indicates a potential advantage of COXE-O and COXE-S as coinitiators in DFR applications.

## 4. Conclusions

Four coumarin ketoxime ester PIs with electron-donating substituents (*tert*iary butyl, methoxy, dimethylamino and methylthio) on the 6-position substituent of the coumarin chromophore were successfully synthesized. The introduction of a *tert*-butyl group is found to remarkably improve the solubility of the coumarin chromophore-based structure. The introduction of heteroatom electron-donating substituents leads to a significant redshift in the light absorption (more than 60 nm); among them, COXE-O and COXE-S exhibit a higher photoinitiation efficiency than that obtained for commercialized PIs OXE-02 and OXE-03 under visible LED irradiation. Moreover, the sensitization of COXEs in the Iod-PF_6_ and BCIM/NPG systems is confirmed, which expands the application of this kind of compound. As a coinitiator, COXEs are successfully used for DRF photolithography. A study of the photolysis mechanism and sensitization mechanism for COXEs indicates that most COXEs can undergo rapid cleavage of the N-O bond and efficiently generate acetoxy radicals or undergo a photoinduced electron transfer reaction between Iod-PF_6_ or NPG upon LED irradiation. In addition, more details about the comparison of 6- and 7-position substituents on coumarin will be disclosed systematically in forthcoming papers.

## Data Availability

Not applicable.

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
