# Peer review of "Coumarin Ketoxime Ester with Electron-Donating Substituents as Photoinitiators and Photosensitizers for Photopolymerization upon UV-Vis LED Irradiation"

_polymers, 2022, doi:10.3390/polym14214588_

Round 1
Reviewer 1 Report
The authors describe a research article entitled “Coumarin Ketoxime Ester with Electron-donating Substituents as Photoinitiators and Photosensitizers for Photopolymerization upon UV‒Vis LED Irradiation”. The topic of the manuscript is interesting, and the manuscript constitutes an interesting article concerning the development of new oxime esters based on coumarins. A short conclusion highlighting the main results of this study is also provided at the end of the document.
The work is well-written and a well-constructed introduction has been established by the authors. Sufficient spectra and figures are included in the manuscript for comprehension and clarity. Interesting and convincing results are also presented in this work. Overall, I think that this is a manuscript that I recommend for publication after inclusion of minor revision.
1) The authors demonstrated that oxime esters could initiate a polymerization. However, concerning the mechanism, evidence of a decarboxylation reaction should be given. Notably, if CO2 is released during polymerization, a peak should be detected at 2337 cm-1.
2) concerning the NMR of the different coumarins, NMR spectra added in SI are too small to be examined. Please increase the size. Additionally, when doublets, triplets are mentioned, coupling constants are missing at present in the text.
3) Did the authors try to initiate the cationic polymerization of epoxides ?
Reviewer 2 Report
This paper describes the 6-position substituted coumarin derivatives as the photoinitiator. While the study should contribute to photo polymerization chemistry, I have some reservations that should be addressed before publication. My comments are as follows.
1. The 6-position substituent on coumarin is not conjugated with the C=N bond that participates in the photocleavage, whereas the 7-position substituent is conjugated. What did the authors expect from the 6 position-substituted derivatives in the photopolymerization?
2. Scheme 2…How are the 6-position substituents involved in photostability and photocleavage?
3. The authors need to compare the 6-position substituted coumarin to the 7-position one for the identical substitute. The comparison should clarify a difference between the 6- and 7-position substituted coumarins in their properties, such as photostability and photo polymerization efficiency.
